# Phytoremediation Potential of Different Genotypes of *Salix alba* and *S. viminalis*

**DOI:** 10.3390/plants13050735

**Published:** 2024-03-05

**Authors:** Jelena Urošević, Dragica Stanković, Dušan Jokanović, Goran Trivan, Aleh Rodzkin, Đorđe Jović, Filip Jovanović

**Affiliations:** 1Electric Power of Serbia, Balkanska 13, 11000 Belgrade, Serbia; urosevicj75@gmail.com; 2Institute for Multidisciplinary Research, University of Belgrade, Kneza Višeslava 1, 11000 Belgrade, Serbia; dstankovic@imsi.bg.ac.rs (D.S.); gorantrivan@gmail.com (G.T.); 3Faculty of Forestry, University of Belgrade, Kneza Višeslava 1, 11000 Belgrade, Serbia; dusan.jokanovic@sfb.bg.ac.rs; 4International Sakharov Environmental Institute, Belarusian State University, Dauhabrodskaja 23/1, 220070 Minsk, Belarus; aleh.rodzkin@rambler.ru; 5Institute of Forestry, Kneza Višeslava 3, 11000 Belgrade, Serbia; jovicdjm@yahoo.com

**Keywords:** *Salix*, clones, soil, heavy metals, phytoremediation, selection

## Abstract

Elevated concentrations of heavy metals result in soil degradation, a reduction in plant yields, and a lower quality of agricultural products, which directly endangers people, animals, and the ecosystem. The potential of three clones of *Salix alba* (347, NS 73/6, and B-44) and one genotype of *S. viminalis* for the phytoextraction of heavy metals was investigated, with the aim of identifying the most physiologically suitable willow genotypes for use in soil phytoremediation. The experiment was placed on the contaminated soil substrate collected in Kolubara Mining Basin (Serbia), enriched by high loads of heavy metal salts, and a control medium. Significant differences in the concentrations of heavy metals were recorded between the contaminated and control plant material, especially when it comes to nickel (Ni), copper (Cu), cadmium (Cd), and lead (Pb), confirming that *S. alba* and *S. viminalis* are hyperaccumulator species of heavy metals. Clone 347 shows the greatest uptake of Cd and chromium (Cr), and clone B-44 takes up these metals only to a lesser extent, while clone NS 73/6 shows a less pronounced uptake of Cr. The roots have the greatest ability to accumulate Ni and Pb, Cu is absorbed by all plant organs, while Cd is absorbed by the leaves. The organ that showed the greatest ability to accumulate heavy metals was the root, which means that willows have a limited power to translocate heavy metals to above-ground organs. The studied genotypes of *S. alba* have a higher potential for the phytostabilization of Cu and Cd, as well as the phytoextraction of Cd, compared with *S. viminalis*. The results confirm the assumption of differences between different willow genotypes in terms of the ability to phytoextract certain heavy metals from soil, which is important information when selecting genotypes for soil phytoremediation.

## 1. Introduction

Chemical elements whose atomic density exceeds 5 g/cm^3^ are classified as heavy metals. Most pollution is caused by metals and metalloids, such as arsenic (As), cadmium (Cd), chromium (Cr), copper (Cu), lead (Pb), nickel (Ni), mercury (Hg), selenium (Se), silver (Ag), zinc (Zn), aluminum (Al), cesium (Cs), cobalt (Co), manganese (Mn), molybdenum (Mo), strontium (Sr), and uranium (U) [1]. Some of them, in smaller concentrations, are necessary for the optimal growth and development of plants (Cu, Zn, Mn, Fe, Ni, Co, and Mo), while others have a harmful effect even at very low concentrations in the soil, water, and atmosphere, which are defined as environmental pollutants (Cd, Pb, Hg, and Cr) [2]. Although the accumulation and concentration of heavy metals in the environment are a consequence of natural lithogenic and pedogenic processes, the largest sources of heavy metals are of anthropogenic origin [3]. Regardless of the origin, elevated concentrations of heavy metals in soil lead to permanent degradation, reduced plant yields, and poorer-quality agricultural products, thus directly endangering humans, animals, and the ecosystem [4,5]. The consequences that can occur due to the presence of heavy metals in the environment depend on the size and type of pollution, as well as synergistic action with other pollutants, but the main problem is that heavy metals cannot be destroyed but only moved to the zones of the biosphere where their impact would be less harmful, that is, transformed and deposited into inactive chemical forms [6].

One approach to soil reclamation is phytoremediation, which is considered an ecological alternative to physico-chemical technologies [6]. According to the U.S. Environmental Protection Agency (EPA) [7], phytoremediation is defined as a technology that uses plants and their rhizospheric microorganisms to remove, degrade, or fix harmful chemical substances present in soil, and undergroundand surface water, as well as in the atmosphere. In many developed countries, phytoremediation has been adopted as a suitable strategy for the reclamation of soil contaminated with heavy metals due to its general public acceptance and the numerous advantages of this method compared with other physico-chemical remediation methods [8]. It is a method that brings the benefit of low investment and maintenance costs compared with other forms of remediation [9]. Phytoremediation, in the form of phytoextraction, was first proposed by Chaney (1983) [10]. Phytoextraction is a technology that removes heavy metals, metalloids, and radionuclides from the environment by using suitable species and genotypes of plants that are capable of absorbing all these pollutants and accumulating them in parts of plant tissues [2]. Such an approach implies using plants that are able to absorb contaminants through the root system and translocate and/or accumulate them to the above-ground parts, stems, and leaves [11,12,13]. The advantage of fast-growing species of deciduous trees, such as species from the genus *Salix* L. (willow), lies precisely in this ability; therefore, they can be used to reclaim habitats polluted by various agents, including heavy metals. 

In the second half of the 20th century, several researchers came up with the idea of using plants in remediation for the extraction of heavy metals [10,14]. After the discovery that some plant species have the ability to accumulate heavy metals, i.e., the hyperaccumulator trait, the interest in phytoremediation as a type of reclamation has grown rapidly [15]. Thus, as a possible phytoremediation crop, willows have been studied since the beginning of the last decade of the 20th century [16,17,18,19,20,21,22,23,24]. Early research already indicated the pronounced ability of these deciduous species to take up and accumulate large amounts of Zn and Cd [16]. One study confirmed that the potential of willows in the phytoextraction of Cd is exceptional [21]. It was shown that the concentrations of Cd, Zn, and Cu in the soil decrease after reclamation with basket willow (*Salix viminalis* L.) [19,23]. In doing so, it was determined that the phytoextraction potential of heavy metals is largely dependent on specific characteristics of willow genotypes, in the sense that certain genotypes of willows can extract more heavy metals from the soil compared with other genotypes of the same species, as indicated by numerous studies [17,19,21,25,26,27,28,29,30].

This research is based on the following assumptions: (1) there is a possibility of using different willow genotypes for the phytoremediation of soil polluted by heavy metals; (2) there are differences among genotypes in terms of the ability to phytoextract certain heavy metals from soil; and (3) there is a potential for selecting the most effective willow genotypes for phytoremediation for plantation cultivation. Consequently, this research examines the potential of four willow genotypes for the phytoextraction of heavy metals (Cd, Pb, Ni, Cu, Cr, and As), their accumulation, translocation, and adaptation to an elevated presence in the soil, with the aim of identifying the most physiologically suitable genotypes for application in the phytoremediation of soils contaminated with heavy metals.

## 2. Results

### 2.1. Soil Analysis 

Based on the determined values of the heavy metal content in the soil sampled from the contaminated area and the control experimental field, an overview of the results of descriptive statistics and ANOVA is presented (Table 1), with a visual representation of the basic statistical parameters (Figure 1).

The variation range of heavy metal concentration in the contaminated soil was 36.47–39.17 mg/kg for Ni, 12.36–18.87 mg/kg for Cu, 0.38–0.49 mg/kg for Cd, 50.89–54.70 mg/kg for Cr, 22.94–29.81 mg/kg for Pb, and 13.73–17.90 mg/kg for As. On the other hand, the variation range in the control soil was 25.85–27.63 mg/kg for Ni, 14.68–17.11 mg/kg for Cu, 0.44–0.53 mg/kg for Cd, 34.94–41.71 mg/kg for Cr, 20.27–26.24 mg/kg for Pb, and 5.89–9.71 mg/kg for As. According to the CV% values, most of the studied variables have a low degree of variability (0–10%), except for Cu and As, whose values of this coefficient were moderate (10–20%) in the contaminated and control soil, respectively. The mean values of heavy metal content in the contaminated soil were 37.60 mg/kg for Ni, 14.14 mg/kg for Cu, 0.42 mg/kg for Cd, 52.69 mg/kg for Cr, 26.96 mg/kg for Pb, and 15.75 mg/kg for As. On the other hand, the mean values in the control soil were 26.80 mg/kg for Ni, 15.90 mg/kg for Cu, 0.48 mg/kg for Cd, 38.90 mg/kg for Cr, 23.71 mg/kg for Pb, and 7.48 mg/kg for As.

As expected, the stated mean values were higher in the contaminated soil compared with the control soil, except for Cu and Cd, for which the reverse was the case. ANOVA showed that there were statistically significant differences (*p* ˂ 0.05) in the mean values of heavy metal concentration in the control and contaminated soil, except for Cu. Most of the variables contribute significantly to the differentiation between the contaminated and control soil, with Ni having the greatest contribution, while Cd and Pb contribute to the least differentiation, as can be seen from the F-ratio values (Table 1; Figure 1).

In comparison with limit values of polluting, harmful, and dangerous substances in soil, according to the “Official Gazette of RS” No. 30/2018 and No. 64/2019 [31], in the analyzed contaminated soil, only the average value of the Ni concentration (37.60 mg/kg) stands out, which was above maximum limit value (MLV) (35 mg/kg) but lower than remediation value (RV) (210 mg/kg). For this reason, during the experiment with plant material, the soil was additionally contaminated with an aqueous solution of salts of heavy metals. 

### 2.2. Analysis of Plant Material

In order to examine the ability of willows to absorb heavy metals, the content of heavy metals in the roots, stems, and leaves of willows grown on the contaminated and control soils were first analyzed. Based on this analysis, the results of descriptive statistics and ANOVA were presented (Table 2), with a visual representation of the basic statistical parameters of the variables with the highest F-ratio (Figure 2). 

The variation range of heavy metal content in the roots of the contaminated plant material was 14.21–27.83 mg/kg for Ni, 25.87–56.67 mg/kg for Cu, 1.46–6.27 mg/kg for Cd, 1.93–8.62 mg/kg for Cr, 3.62–7.86 mg/kg for Pb, and 0.55–11.65 mg/kg for As. On the other hand, the variation range in the roots of the control plant material was 1.06–4.40 mg/kg for Ni, 5.85–10.56 mg/kg for Cu, 0.24–0.47 mg/kg for Cd, 0.43–6.17 mg/kg for Cr, 0.00–1.99 mg/kg for Pb, and 0.00–0.00 mg/kg for As. The variation range in the stems of the contaminated plant material was 0.00–5.59 mg/kg for Ni, 6.51–10.45 mg/kg for Cu, 1.80–5.62 mg/kg for Cd, 1.43–5.78 mg/kg for Cr, 0.00–2.70 mg/kg for Pb, and 0.00–3.76 mg/kg for As. On the other hand, the variation range in the stems of the control plant material was 0.00–1.32 mg/kg for Ni, 3.18–6.01 mg/kg for Cu, 0.49–0.60 mg/kg for Cd, 0.00–2.96 mg/kg for Cr, 0.00–0.00 mg/kg for Pb, and 0.00–0.01 mg/kg for As. The variation range in the leaves of the contaminated plant material was 0.00–6.09 mg/kg for Ni, 11.78–17.84 mg/kg for Cu, 2.88–8.09 mg/kg for Cd, 0.00–5.47 mg/kg for Cr, 0.00–1.87 mg/kg for Pb, and 0.00–4.78 mg/kg for As. On the other hand, the variation range in the leaves of the control plant material was 0.00–3.40 mg/kg for Ni, 4.23–6.43 mg/kg for Cu, 0.80–1.42 mg/kg for Cd, 0.00–1.86 mg/kg for Cr, 0.00–0.00 mg/kg for Pb, and 0.00–0.00 mg/kg for As. CV% values ranged from low (˂10%) to very high variability (>60%) depending on the heavy metal, plant organ, and contamination. The mean values of heavy metal content in the roots of the contaminated plant material were 21.26 mg/kg for Ni, 45.60 mg/kg for Cu, 3.37 mg/kg for Cd, 5.01 mg/kg for Cr, 5.08 mg/kg for Pb, and 5.87 mg/kg for As. On the other hand, the mean values in the roots of the control plant material were 2.87 mg/kg for Ni, 8.29 mg/kg for Cu, 0.35 mg/kg for Cd, 3.17 mg/kg for Cr, 0.43 mg/kg for Pb, and 0.00 mg/kg for As. The mean values in the stems of the contaminated plant material were 1.49 mg/kg for Ni, 8.34 mg/kg for Cu, 3.32 mg/kg for Cd, 3.18 mg/kg for Cr, 0.67 mg/kg for Pb, and 1.29 mg/kg for As. On the other hand, the mean values in the stems of the control plant material were 0.38 mg/kg for Ni, 4.73 mg/kg for Cu, 0.54 mg/kg for Cd, 1.41 mg/kg for Cr, 0.00 mg/kg for Pb, and 0.00 mg/kg for As. Finally, the mean values in the leaves of the contaminated plant material were 2.24 mg/kg for Ni, 13.98 mg/kg for Cu, 4.79 mg/kg for Cd, 1.30 mg/kg for Cr, 0.29 mg/kg for Pb, and 0.71 mg/kg for As. In contrast, the mean values in the leaves of the control plant material were 0.92 mg/kg for Ni, 5.26 mg/kg for Cu, 1.06 mg/kg for Cd, 0.22 mg/kg for Cr, 0.00 mg/kg for Pb, and 0.00 mg/kg for As.

As expected, the mean values were higher in the contaminated than in the control plant material. ANOVA showed that there are statistically significant differences (*p* ˂ 0.05) in the mean values of the heavy metal contents in plant organs of the control and contaminated plant materials (Table 2), which supports the assumption of the ability of willows to take up heavy metals. All variables contribute to the differentiation between the control and contaminated plant materials, with the greatest contribution being Ni in the root (Figure 2a), Cu in the root, stem, and leaf (Figure 2b–d), Cd in the leaf (Figure 2e), and Pb in the root (Figure 2f), while Ni in the stem and Cr, Pb, and As in the leaf contribute the least to the differentiation, as can be seen from the values of the F-ratio (Table 2).

The differences between the studied genotypes and the specificity of willow organs in the accumulation of heavy metals were established by descriptive, univariate (ANOVA), and multivariate statistical analyses (CDA) (Table 3) and are presented with CDA scatterplots (Figure 3).

The Ni concentration was recorded in the range of 0.00 mg/kg (clones 2, 3, and 4) to 27.83 mg/kg (clone 4) in the contaminated plant material, as well as from 0.00 mg/kg (clones 1–4) up to 4.40 mg/kg (clone 4) in the material from the control experimental field. The content of Cu was recorded in the range of 6.51 mg/kg (clone 2) to 56.67 mg/kg (clone 3) in the contaminated plant material, as well as from 3.18 mg/kg to 10.56 mg/kg (both values for clone 3) in the material from the control experimental field. The Cd content varied from 1.46 mg/kg (clone 1) to 8.09 mg/kg (clone 4) in the contaminated plant material, as well as from 0.24 mg/kg to 1.42 mg/kg (both values for clone 1) in the material from the control experimental field. The Cr content was recorded in the range of 0.00 mg/kg (clones 1–4) to 8.62 mg/kg (clone 4) in the contaminated plant material, as well as from 0.00 mg/kg (clones 1–4) to 6.17 mg/kg (clone 4) in the control experimental field. The Pb content was recorded in the range of 0.00 mg/kg (clones 1–4) to 7.86 mg/kg (clone 2) in the contaminated plant material, as well as from 0.00 mg/kg (clones 1–4) to 1.99 mg/kg (clone 4) in the control experimental field. The As content varied from 0.00 mg/kg (clones 1–4) to 11.65 mg/kg (clone 1) in the contaminated plant material, as well as from 0.00 mg/kg (clones 1–4) to 0.01 mg/kg (clone 4) in the control experimental field. The CV% values of the studied variables ranged from a low (˂10%) to a very high variability level (>60%) depending on the heavy metal, plant organ, and contamination. The mean values of the heavy metal contents in the contaminated plant material showed a range of values from 6.68 mg/kg (clone 3) to 10.72 mg/kg (clone 4) for Ni, from 19.68 mg/kg (clone 1) to 24.47 mg/kg (clone 2) for Cu, from 2.84 mg/kg (clone 3) to 5.38 mg/kg (clone 4) for Cd, from 2.17 mg/kg (clone 1) to 4.28 mg/kg (clone 4) for Cr, from 1.96 mg/kg (clone 3) to 2.25 mg/kg (clone 4) for Pb, and from 1.79 mg/kg (clone 3) to 3.77 mg/kg (clone 1) for As. On the other hand, the mean values of the heavy metal contents in the control plant material showed a range of values from 1.07 mg/kg (clone 3) to 1.59 mg/kg (clone 4) for Ni, from 6.00 mg/kg (clone 4) to 6.19 mg/kg (clone 1) for Cu, from 0.63 mg/kg (clone 3) to 0.73 mg/kg (clone 2) for Cd, from 1.17 mg/ kg (clone 2) to 2.76 mg/kg (clone 4) for Cr, from 0.00 mg/kg (clone 3) to 0.38 mg/kg (clone 2) for Pb, and from 0.00 mg/kg to 0.00 mg/kg (clones 1–4) for As. ANOVA showed that there were statistically significant differences (*p* < 0.05) between the mean values of the Cd and Cr contents in the contaminated plant material and Cr in the plant material from the control experimental field, with a significant contribution of these variables to the general differentiation of genotypes, which can be seen from the F-ratio values. According to the homogeneous groups in the LSD test, Cd and Cr accumulate to the greatest extent in the plant material of clone 4 and to a lesser extent in the plant material of clone 2; clone 3 also had a relatively pronounced ability to accumulate Cr, but less than clone 4 (Table 3).

In the CDA, the first function accounts for 84.97% of discrimination, while the second function accounts for an additional 11.30% for the contaminated plant material (Table 3; Figure 3a). On the other hand, in the CDA for the control plant material, the first function accounts for 67.08% of the discrimination, and the second function accounts for an additional 16.67% (Table 3; Figure 3b). When it comes to contaminated plant material, the variable Cu with a discriminant value > 0.70 is responsible for the differentiation along the first discriminant axis (DA1), and the variable Cd had an impact on the second axis (DA2). Similarly, in the case of the control material, the variable Cd with a discriminant value > 0.70 was responsible for the differentiation along the DA1 axis, while the variable Ni had an effect on DA2. Therefore, similarly to the results obtained from ANOVA, in the first case, the samples are separated based on the contents of Cu and Cd, and in the second case, based on the contents of Cd and Ni. In both cases, the result of the CDA was three entities differentiated along DA1; for the control material, the entities were separated according to which of the three analyzed plant organs they refer to (root, stem, and leaf) (Figure 3b), while the contaminated material entities were differentiated according to whether they refer to above-ground (stem and leaf) or underground organs (root) (Figure 3a). Thus, for the control material, the leaf samples of the studied genotypes were separated from the other samples on the positive part of DA1, while the root samples were separated on the negative part of the same axis, and a separation of clone 2 root samples along DA2 was observed (Figure 3b), which indicates significant physiological differences between the analyzed genotypes and willow organs regarding the ability to accumulate Cd. On the other hand, in the case of the contaminated material, the root samples of clone 1 and the other genotypes were separated from the other samples on the positive part of DA1, while the stem and leaf samples were separated on the negative part of the same axis, and a trend in the separation of these samples along DA2 according to genotype was observed (Figure 3a), indicating the existence of significant physiological differences between the analyzed organs and willow genotypes in terms of Cu and Cd accumulation. More specifically, the results of this analysis suggest that for the control material of the studied willow genotypes and organs, the leaves have the greatest ability to accumulate Cd, while for the contaminated plant material, the ability to accumulate Cu is observed in the roots of clones 2, 3, and 4, and Cd—in the roots of clone 2 and in the stem and leaves of clone 4.

## 3. Discussion

Soil contaminated with heavy metals threatens the global ecosystem and human health [32]. The elimination of specific heavy metals from the soil has emerged as a foremost priority, presenting a formidable challenge for the scientific community to discover a suitable technique for extracting potentially hazardous substances from the soil while minimizing expenses. Namely, price is the primary factor that determines the search for alternative remediation technologies, such as phytoremediation [33,34]. In principle, the concept of phytoremediation provides a green, cheap, safe, and sustainable solution for the remediation of heavy metal toxicity in soil. However, in order to establish the effectiveness and safety of phytoremediation for a specific site, phytoremediation must be studied on a case-by-case basis, as performed for other forms of remediation. It is necessary that the phytoremediation procedure be preceded by the consideration of factors, such as soil properties, the presence and type of pollutants, climate, type of plants, etc. In addition, it is necessary to consider new strategies for the improvement of phytoremediation, such as different methods and biotechnological techniques, the application of plants in different approaches to phytoremediation, and other factors of possible influence on phytoremediation, which creates opportunities for establishing new strategies in phytoremediation [8].

The selection of plant species to be applied in phytoremediation is an important step that determines the success of phytoremediation [35]. Species from the genus *Salix* belong to the group of hyperaccumulator species and are used in phytoremediation due to their fast growth and high growth rate. Moreover, compared with some other economically significant crops (poplar, sunflower, and tobacco), willows convincingly showed the highest potential for absorption of heavy metals on agricultural land [36]. Therefore, this article examined the potential of certain willow genotypes for the phytoextraction of heavy metals, and the most physiologically suitable genotypes for use in the phytoremediation of soil contaminated with heavy metals were selected.

As expected, the examination of the concentration of heavy metals in the soil sampled from the contaminated site and intended for setting up the experiment with contaminated plant material, as well as in the soil on which the control plant material was grown, revealed higher mean values of these metals in the contaminated soil compared with the control site. The only exception pertains to the concentrations of Cu and Cd. In the control experimental field, these concentrations were higher compared with the soil obtained from the contaminated site. However, these variances were not deemed statistically significant for Cu. The presence of Cd in the control soil can be attributed to the historical anthropogenic impact on the soil where the control plant material was cultivated. The geological weathering of rocks stands as the primary natural source of Cd in soil, as noted in previous studies [37,38]. Conversely, primary anthropogenic sources contributing to Cd include agrochemicals, manufacturing, vehicle emissions, irrigation wastewater, smelting, and mining, as identified in previous research [39,40]. The mean concentration of Cd exhibited a statistically significant elevation in the control soil compared with the contaminated soil. However, Cd was among the variables that contributed the least to the distinction among the soil samples, with the average content in the control soil remaining within the prescribed limits [31].

In accordance with the relevant regulation [31], only the average Ni concentration exceeded the MLV in the contaminated soil. It is known that the average content of this heavy metal in soils around the world amounts to 40 mg/kg [41], which approximately corresponds to the established value. Ni, as a pollutant, is released into the environment through human activities, such as mining, fossil fuel combustion, fertilization, and municipal and industrial wastes, including smelting and electroplating [42,43,44,45], which is in agreement with the use of the land where the soil samples were taken for the purposes of this research. Since the content of Ni in this research is in the intermediate values of MLV and RV, and the contents of other metals were recorded within the permitted range, during the experiment, the soil was additionally contaminated with an aqueous solution of salts of heavy metals. 

The mean values of the heavy metal contents were significantly higher in the contaminated plant material than in the control plant material, which supports the assumption that willows are hyperaccumulators of heavy metals. The biggest differences in the heavy metal contents between the control and contaminated plant materials were observed for Ni, Cu, Cd, and Pb. Although Ni is an essential trace element for the growth and development of higher plants [46], its high concentrations are toxic to plants [47]. According to the literature data [48,49], the average Ni content in plants ranges from 0.1 to 5.0 ppm of dry matter, while for sensitive plants, the toxic values of Ni are higher than 10 ppm, and for tolerant species, they can be higher than 50 ppm [50,51]. The limit values of Ni toxicity in leaves are 25–50 mg/kg of dry plant matter [52,53]. Comparing the established Ni content in the contaminated willow material with the literature data, it can be concluded that this content is below the critical level for plants. Cu is a trace element of essential nature for plants, which affects the metabolism of nitrogenous compounds and carbohydrates, the formation and fertility of pollen, and the resistance of plants to diseases [54]. Ross (1994) [55] stated that concentrations of this heavy metal in contaminated plants are 20–100 mg/kg. The recorded concentration of Cu in the contaminated willow material was within the range of values stated in the literature. In contrast to Ni and Cu, Cd and Pb represent toxic elements for plants. Controlling Cd accumulation in plants is demanding, as most essential nutrient transporters, such as Cu, Mn, Fe, and Zn, also facilitate its absorption [56]. Lux et al. (2002) and Lunackovà et al. (2003) [57,58] reported that the amount of Cd taken up by *Salix alba* L. ranged from 2.0 to 62.1 μg/g of dry matter in the stem and 3.0 to 160 μg/g in leaves and shoots. Borišev et al. (2009) [59] pointed out that Cd concentrations in the leaves of species *S. alba* (476.6 μg/g of dry matter), *S. matsudana* Koidz. (389.0 μg/g), and *S. nigra* L. (507.5 μg/g), as well as in the shoots of these three species (338.9 μg/g, 259.6 μg/g, and 347.0 μg/g, respectively), rank among the highest recorded concentrations of heavy metals accumulated in willows in aquatic culture. The Cd content found in our research was lower than the values reported in the literature because the plant material was grown on a soil substrate, not in hydroculture. In addition, the Pb content was lower than the value considered harmful to forest trees (>15 μg/g) [60].

The phytoextraction potential of heavy metals largely depends on the genotypic specificity of willows, as indicated by a large number of studies [17,19,21,25,26,27,28,29,30]. The results obtained in this article also indicate the specificities of willow genotypes in the accumulation of heavy metals. This finding primarily refers to Cd and Cr, which, compared with the other genotypes, were accumulated to the greatest extent by clone 4 (347) and to a somewhat lesser extent by clone 2 (B-44), while clone 3 (NS 73/6) of white willow (*S. alba*) also had a pronounced ability to accumulate Cr, in contrast to an unnamed clone of the basket willow (clone 1). In the study by Mleczek et al. (2009) [22], the highest concentration of Cr was recorded with *S. petiolaria* “Rigida” (3.06 mg/kg and 3.24 mg/kg) and *S. japonica* Thunb. (2.79 mg/kg and 2.95 mg/kg), while the lowest concentration was observed in *S. purpurea* var. *angustifolia* Kerner (0.44 mg/kg and 0.61 mg/kg) and *S. purpurea* var. Schultze Schultze (0.71 mg/kg and 0.83 mg/kg). In other research studies, it was shown that the concentrations of Cd, Zn, and Cu in the soil decreased after reclamation with basket willow [19,23]. This willow species reportedly reduces the total Cd concentration in agricultural land by up to 25% after only four years of reclamation [23]. Landberg and Greger (2022) [61] pointed out that during soil treatment with this species, over a period of 10 years, the concentration of many metals and metalloids was reduced by 21–87% (Cr, 21%; As, 30%; Cd, 54%; Zn, 61%; Cu, 62%; Pb, 63%; and Ni, 87%). The differences in bioaccumulation among different clones are genotype-specific, i.e., affected more by the (physiological) properties of clones than by the soil element concentrations or soil properties [62]. For this reason, if there is soil that requires effective remediation, it is possible to select appropriate willow genotypes that are selective for the targeted pollutant.

The success of phytoremediation depends on the degree of soil pollution, the availability of metals for root uptake (bioavailability), and the ability of plants to absorb and accumulate heavy metals in their organs [63]. When it comes to the uptake of heavy metals in the organs of the examined plant material, the highest accumulation of Ni and Pb was observed in the roots, Cu in all organs, and Cd in the leaves. According to the literature, Cu is mainly deposited in the leaves, where it affects the ability to retain water [64]. However, when its concentration in the external environment is high, it is largely accumulated in the roots of certain plant species [65]. Thus, the elevated concentration of Cu in the soil increased its content in the roots of *S. jiangsuensis* CL “J-172” and *S. babylonica* Linn [66]. This is considered to be one of the mechanisms of plant protection against excess Cu [65]. On the other hand, Cd belongs to the group of heavy metals that are easily absorbed by plants through the roots, but due to its easy mobility, it is also translocated to the leaves [67,68,69,70]. Tőzsér et al. (2017) [71] noted that the accumulation of this heavy metal differed significantly among organs and stated that willows are prosperous accumulators of Cd in above-ground organs. In general, in relation to the studied plant materials, the root was singled out as the organ with the greatest capacity to accumulate heavy metals, which confirms the results of previous research on woody species [68,72]. A number of hyperaccumulator species have a limited ability to translocate heavy metals into their above-ground parts [14,73,74], which Baker (1981) [14] and Stoltz and Greger (2002) [75] explained as a protective mechanism of plants that protects the photosynthetic apparatus from the negative effects of heavy metals.

The study of phytoremediation plants can contribute to the development of new strategies to mitigate the negative effects of heavy metal contamination and the need for research, monitoring, and strategies to mitigate these environmental problems in order to preserve the well-being of the environment and mankind [15]. Taking into consideration the results of the CDA for the organs of the studied willow genotypes, it was observed that Cd was accumulated the most in the roots of clone B-44 and in the stem and leaves of clone 347, and Cu was primarily accumulated in the roots of clones B-44, NS 73/6, and 347. This represents information that should be taken into account when selecting genotypes for the phytoremediation of soils contaminated with these heavy metals. The successful application of phytoextraction depends on factors that include the degree of soil or water pollution, the biological availability of metals for uptake by the roots of a particular plant, and the plant’s potential to adopt and accumulate heavy metals in plant organs that are easy to remove from the polluted habitat [2]. The observation that the root exhibits greater activity than the above-ground organs in the uptake of certain heavy metals underlines the specific and dynamic nature of the response of plants to exposure to high concentrations of heavy metals, a better understanding of which can contribute to the development of effective and sustainable strategies for the remediation of environments polluted by heavy metals [15].

## 4. Materials and Methods

### 4.1. Soil Sampling and Analysis

The soil for the experiment (about 5 tons of soil) was excavated with an excavator from Kolubara Mining Basin at the “Kanal Crnih voda” site, Vreoci, from different depths (Figure 4a), then homogenized and packed in polyethylene bags (of 10 l). Exactly 120 bags per clone were filled with this soil (480 bags in total). Bags with soil were transferred to the Nursery of the Faculty of Forestry, University of Belgrade, and were used to plant four willow clones. At the beginning of each vegetation period, the substrate was watered with an aqueous solution of heavy metal salts [Cd(NO_3_)_2_, 112.4 ppm; CuSO_4_·5H_2_O, 63.5 ppm; K_2_Cr_2_O_7_, 104.0 ppm; Na_2_HAsO_4_·7H_2_O, 74.9 ppm; NiCl_2_·6H_2_O, 58.7 ppm; PbNO_3_, 207.2 ppm] in concentrations of 10–3 mol/dm^3^ to the optimal water capacity (65–75%). The criterion for selecting these metals was their toxic effect on plants, as well as the ability of willows to absorb them from the soil, as indicated by data in the literature. For laboratory analyses, the soil was sampled by taking a few grams of substrate from each bag. At the same time, soil samples were taken from the nursery of the Faculty of Forestry (Figure 4b), where the control plant material was grown. 

The soil samples were dried to an air-dry state and then manually ground in a mortar and sieved through a 2 mm sieve. For the samples prepared in this way, the hydroscopic moisture content was determined by drying to an oven-dry state at 105 °C. Chemical analyses were performed on the air-dried samples, and the amounts of determined analytes were recalculated to the oven-dried state of the soil. Heavy metals, dissolved in aqua regia, were extracted by digestion in a microwave digestion platform ETHOS EASY (Milestone Srl, Sorisole, Italy). The resulting extract was filtered and filled up to 50 mL. For the obtained extract, in aqua regia, the content of heavy metals was measured in several repetitions for contaminated and control soil (n = 6) in an ICP-OES spectrometer VISTA-PRO (Varian Australia Pty. Ltd., Victoria, Australia), according to the methodology of Cools and De Vos (2020) [76]. The values obtained in this way were compared with the maximum limit values (MLVs), as well as the remediation values (RVs), in accordance with the regulation on limit values of polluting, harmful, and dangerous substances in soil [31]. 

### 4.2. Experiment, Sampling, and Analysis of Plant Material

The experiment of the reaction of four willow genotypes for the presence of heavy metals in uncontaminated and contaminated soil and their phytoremediation potential was set up in the Nursery of the Faculty of Forestry, University of Belgrade. The cuttings of the clones (Figure 4c) were obtained from the nursery of PE “Vojvodina Šume”, Kać, where the plant material included a clone of the basket willow and three clones of the white willow (347, NS 73/6, and B-44), which are referred to in the text as “clone 1”, “clone 2”, “clone 3”, and “clone 4”, respectively. At the end of March 2019, three cuttings were planted in each bag with the contaminated substrate (Figure 4d). The remaining cuttings (1440) were planted in the control plot (nursery bed) (Figure 4e) to examine the differences in the influence of the contaminated and non-contaminated substrates. The willow cuttings were soaked in fungicide [Cu_2_(OH)_3_Cl] before planting.

For the purpose of heavy metal analysis, the seedlings were cut at the end of the vegetation period in 2021. To determine the content of heavy metals in the contaminated and control plant materials, root, stem, and leaf samples were taken. The samples were dried at a temperature of 40 °C and then ground into powder. The moisture content of the prepared samples was determined by drying at 105 °C. The quantity of 0.2 g of ground material of known humidity was filled with 8 mL of concentrated nitric acid (HNO_3_) and 2 mL of hydrogen peroxide (H_2_O_2_). Digestion was performed in a microwave digestion platform ETHOS EASY (Milestone Srl, Sorisole, Italy). The crushed samples were brought up to 25 mL. The determination of total amounts of heavy metals in the plant materials was performed in several repetitions per organ and clone (n = 6) in an ICP-OES spectrometer VISTA-PRO (Varian Australia Pty. Ltd., Victoria, Australia) (Figure 4f), according to the methodology of Rautio et al. (2016) [77].

### 4.3. Statistical Analysis

The statistical analysis included more than 900 numerical data points, which were processed in accordance with descriptive, univariate, and multivariate statistical methods. For each variable, the following basic parameters were determined: minimum value (MIN), maximum value (MAX), mean value ($\overset{–}{X}$), standard deviation (SD), and coefficient of variation (CV%). The significance of differences among mean values was determined by analysis of variance (ANOVA) and the post hoc Fisher’s least significant difference (LSD) test. The presence of heavy metals in the plant material was also analyzed using canonical discriminant analysis (CDA) in order to identify the variables that contribute the most to the discrimination of samples of genotypes and organs. Statistical analyses were preceded by testing the normality and collinearity of the data. Outliers were excluded. Statistical analyses were performed in Statgraphics Centurion ver. XVI.I (2009; Statpoint Technologies, Inc., Warrenton, VA, USA).

## 5. Conclusions

Based on the results obtained by researching the potential of certain white and basket willow genotypes for the phytoextraction of heavy metals, it is possible to draw the following conclusions:−White willow and basket willow are hyperaccumulator species of heavy metals, bearing in mind the significant differences in the concentration of heavy metals [especially nickel (Ni), copper (Cu), cadmium (Cd), and lead (Pb)] between the contaminated and control plant materials; −The degree of accumulation of heavy metals by willows depends on the genotype, and there is a specificity of willow genotypes in the phytoextraction of heavy metals from the soil due to their physiological differences. Thus, it is possible to identify genotypes that are selective for the targeted pollutant and applied in the phytoremediation of soil contaminated with certain heavy metals. Compared with the basket willow, Cd and chromium (Cr) are absorbed to the greatest extent by clone 347 of the white willow and to a somewhat lesser extent by clone B-44, while clone NS 73/6 of the same species shows a less pronounced ability to accumulate Cr;−There is a specificity of willow genotypes in the phytoextraction of heavy metals and their accumulation in different plant organs. Roots the greatest ability to accumulate Ni and Pb, Cu is absorbed by all plant organs, while Cd is absorbed by the leaves. The organ that stands out for the greatest ability to accumulate heavy metals is the root, which means that willows have a limited power to translocate heavy metals to above-ground organs. The reason for the good survival of willows on soil contaminated with heavy metals lies precisely in the retention of heavy metals in the roots and their relatively weak translocation into the assimilation organs;−Absorption of Cu in the root (phytostabilization) depends on the genotype of the willow, whereby clones 347, NS 73/6, and B-44 of the white willow have a greater phytostabilization potential of this heavy metal compared with the basket willow clones;−White willow clone NS 73/6 has the highest Cd phytostabilization ability, and the B-44 clone of the same species has the ability to phytoextract (accumulate in above-ground organs), which means that the white willow has a higher Cd accumulation potential than the basket willow.

These research results contribute to the knowledge of the possibility of using different willow genotypes for soil phytoremediation and support the assumption that there are differences among willow genotypes in the ability to phytoextract certain heavy metals from the soil, which is information that should be taken into account when choosing willow genotypes for the phytoremediation of soil contaminated with heavy metals.

## Figures and Tables

**Figure 1 plants-13-00735-f001:**
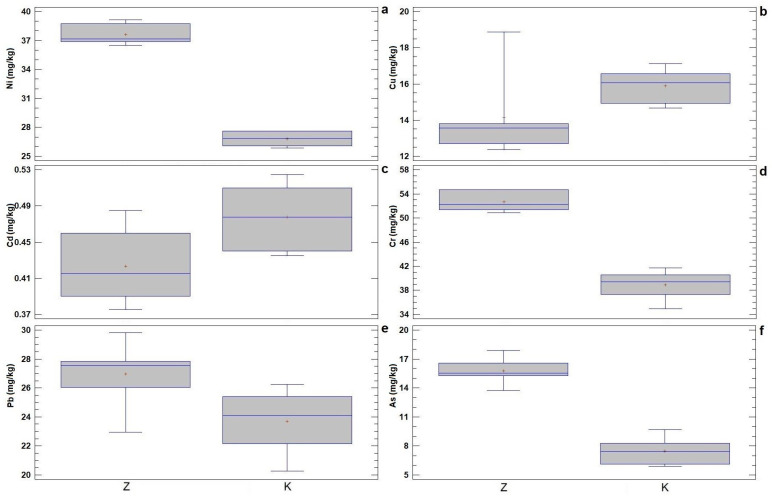
Box-and-whisker plots of the basic statistical parameters for the content of heavy metals (mg/kg) in the contaminated soil (Z) and in the control experimental field (K): (**a**) Ni; (**b**) Cu; (**c**) Cd; (**d**) Cr; (**e**) Pb; and (**f**) As. Legend: middle sign—mean value, middle line—median, box—mean value and standard deviation, whiskers—variation range.

**Figure 2 plants-13-00735-f002:**
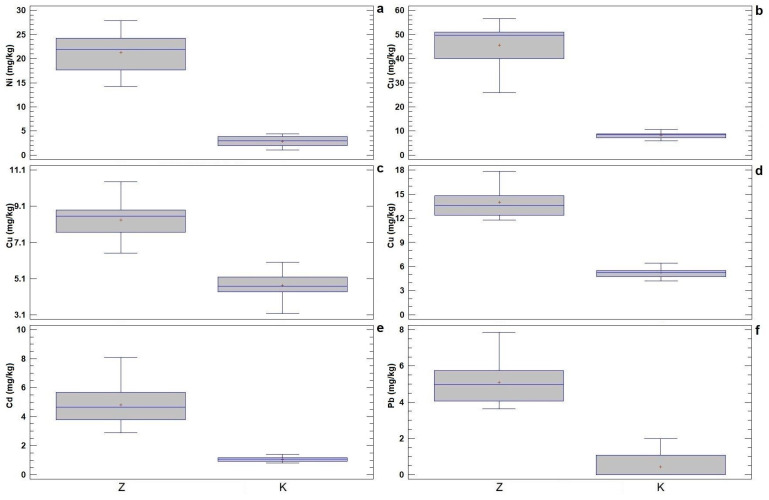
Box-and-whisker plots of basic statistical parameters for the content of heavy metals (mg/kg) in the roots, stems, and leaves of willows on the contaminated soil (Z) and the control experimental field (K). Variables with the highest F-ratio values: (**a**) Ni in the roots; (**b**) Cu in the roots; (**c**) Cu in the stems; (**d**) Cu in the leaves; (**e**) Cd in the leaves; and (**f**) Pb in the roots. Legend: middle sign—mean value, middle line—median, box—mean value and standard deviation, whiskers—variation range.

**Figure 3 plants-13-00735-f003:**
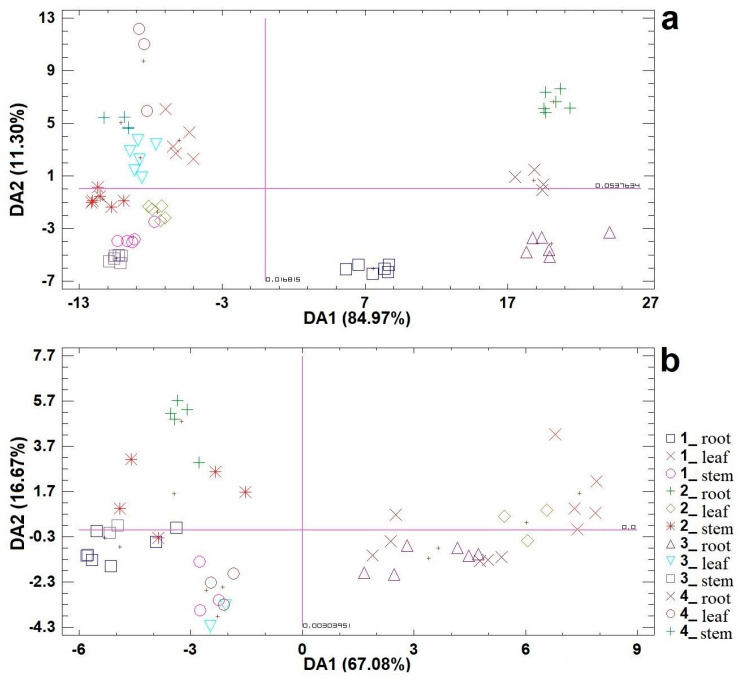
Canonical discriminant analysis (CDA) scatterplots for the content of heavy metals (mg/kg) in different organs (root, leaf, and stem) of four willow genotypes (1–4): (**a**) on contaminated soil and (**b**) on the control experimental field.

**Figure 4 plants-13-00735-f004:**
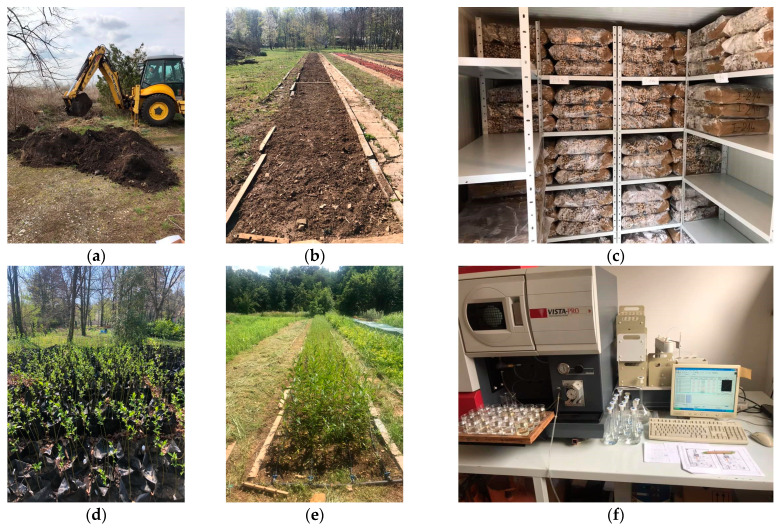
Experiment setup. (**a**) Excavating contaminated soil from Kolubara Mining Basin, Serbia. (**b**) Control plot (nursery bed) in the Nursery of the Faculty of Forestry, University of Belgrade. (**c**) Cuttings of the willow clones used for the experiment. (**d**) Polyethylene bags filled with the contaminated soil and used to plant the willow clones. (**e**) Willow clones planted on the control plot (nursery bed). (**f**) Determination of the heavy metal content in plant samples with an ICP-OES spectrometer VISTA-PRO.

**Table 1 plants-13-00735-t001:** Descriptive statistics ($\overset{–}{X}$—mean value, MIN—minimum value, MAX—maximum value, SD—standard deviation, CV%—coefficient of variation) and analysis of variance (ANOVA) for the content of heavy metals (mg/kg) in the contaminated soil and in the control experimental field.

Soil		Ni	Cu	Cd	Cr	Pb	As
Contaminated	$\overset{–}{X}$	37.60	14.14	0.42	52.69	26.96	15.75
	MIN	36.47	12.36	0.38	50.89	22.94	13.73
	MAX	39.17	18.87	0.49	54.70	29.81	17.90
	SD	1.09	2.39	0.04	1.69	2.31	1.40
	CV%	2.91	16.87	9.98	3.21	8.58	8.88
Control	$\overset{–}{X}$	26.80	15.90	0.48	38.90	23.71	7.48
	MIN	25.85	14.68	0.44	34.94	20.27	5.89
	MAX	27.63	17.11	0.53	41.71	26.24	9.71
	SD	0.82	0.94	0.04	2.45	2.23	1.44
	CV%	3.08	5.89	7.57	6.30	9.39	19.33
ANOVA	F	372.90	2.83	5.69	128.75	6.15	101.66
	*p* ^1^	**0.0000**	0.1232	**0.0383**	**0.0000**	**0.0326**	**0.0000**

^1^ Variables with *p* ˂ 0.05 (ANOVA) are marked with bold numbers.

**Table 2 plants-13-00735-t002:** Descriptive statistics ($\overset{–}{X}$—mean value, MIN—minimum value, MAX—maximum value, SD—standard deviation, CV%—coefficient of variation) and analysis of variance (ANOVA) for the content of heavy metals (mg/kg) in the roots (R), stems (S), and leaves (L) of willows in the contaminated soil and in the control experimental field.

Plant Material		Ni			Cu			Cd			Cr			Pb			As		
		R	S	L	R	S	L	R	S	L	R	S	L	R	S	L	R	S	L
Contaminated	$\overset{–}{X}$	21.26	1.49	2.24	45.60	8.34	13.98	3.37	3.32	4.79	5.01	3.18	1.30	5.08	0.67	0.29	5.87	1.29	0.71
	MIN	14.21	0.00	0.00	25.87	6.51	11.78	1.46	1.80	2.88	1.93	1.43	0.00	3.62	0.00	0.00	0.55	0.00	0.00
	MAX	27.83	5.59	6.09	56.67	10.45	17.84	6.27	5.62	8.09	8.62	5.78	5.47	7.86	2.70	1.87	11.65	3.76	4.78
	SD	4.05	2.01	1.72	9.32	0.89	1.93	1.73	1.42	1.43	1.76	1.27	1.68	1.10	1.00	0.53	3.56	1.61	1.33
	CV%	19.06	134.69	76.84	20.43	10.71	13.81	51.27	42.76	29.87	35.23	40.02	129.16	21.64	150.08	183.32	60.73	125.41	185.84
Control	$\overset{–}{X}$	2.87	0.38	0.92	8.29	4.73	5.26	0.35	0.54	1.06	3.17	1.41	0.22	0.43	0.00	0.00	0.00	0.00	0.00
	MIN	1.06	0.00	0.00	5.85	3.18	4.23	0.24	0.49	0.80	0.43	0.00	0.00	0.00	0.00	0.00	0.00	0.00	0.00
	MAX	4.40	1.32	3.40	10.56	6.01	6.43	0.47	0.60	1.42	6.17	2.96	1.86	1.99	0.00	0.00	0.00	0.01	0.00
	SD	1.03	0.48	1.00	1.24	0.68	0.52	0.06	0.03	0.16	1.68	0.67	0.48	0.69	0.00	0.00	0.00	0.00	0.00
	CV%	35.80	126.27	108.16	14.98	14.29	9.97	18.46	6.36	15.06	53.07	47.66	215.53	161.00	0.00	00.00	0.00	424.26	0.00
ANOVA	F	389.89	6.66	10.04	378.44	239.50	436.49	66.95	68.63	161.13	13.01	33.57	8.40	297.50	9.75	6.85	62.30	11.37	6.95
	*p* ^1^	**0.0000**	**0.0132**	**0.0028**	**0.0000**	**0.0000**	**0.0000**	**0.0000**	**0.0000**	**0.0000**	**0.0008**	**0.0000**	**0.0059**	**0.0000**	**0.0032**	**0.0122**	**0.0000**	**0.0017**	**0.0114**

^1^ Variables with *p* ˂ 0.05 (ANOVA) are marked with bold numbers.

**Table 3 plants-13-00735-t003:** Descriptive statistics ($\overset{–}{X}$—mean value, MIN—minimum value, MAX—maximum value, SD—standard deviation, CV%—coefficient of variation), analysis of variance (ANOVA), and canonical discriminant analysis (CDA) for heavy metal content (mg/kg) in the plant material of four willow genotypes (clones 1–4) on the contaminated soil (Z) and the control experimental field (K).

Genotype						Ni		Cu		Cd		Cr		Pb		As	
						Z	K	Z	K	Z	K	Z	K	Z	K	Z	K
Clone 1				$\overset{–}{X}$ ^1^		7.33 a	1.22 a	19.68 a	6.19 a	2.94 c	0.70 a	2.17 b	1.39 b	2.03 a	0.04 b	3.77 a	0.00 a
				MIN		0.03	0.00	7.86	4.23	1.46	0.24	0.00	0.00	0.00	0.00	0.00	0.00
				MAX		22.49	3.40	32.71	10.03	5.72	1.42	4.10	3.03	6.73	0.71	11.65	0.00
				SD		7.85	0.87	9.23	1.81	1.46	0.42	1.13	1.09	2.31	0.17	4.29	0.00
				CV%		107.13	71.38	46.92	29.20	49.50	59.99	52.00	78.06	113.75	424.26	113.71	0.00
Clone 2				$\overset{–}{X}$		8.98 a	1.46 a	24.47 a	6.16 a	4.15 b	0.73 a	3.55 ab	1.17 b	2.03 a	0.38 a	1.84 a	0.00 a
				MIN		0.00	0.00	6.51	3.94	2.88	0.35	0.00	0.00	0.00	0.00	0.00	0.00
				MAX		25.78	3.83	53.79	9.29	6.27	1.19	6.17	2.97	7.86	1.82	4.62	0.00
				SD		11.09	1.39	19.99	1.85	1.28	0.39	1.95	0.94	2.89	0.63	1.81	0.00
				CV%		123.52	95.05	81.71	30.05	30.79	53.31	54.88	80.50	142.07	165.43	98.53	0.00
Clone 3				$\overset{–}{X}$		6.68 a	1.07 a	24.13 a	6.14 a	2.84 c	0.63 a	2.86 ab	1.26 b	1.96 a	0.00 b	1.79 a	0.00 a
				MIN		0.00	0.00	7.94	3.18	1.80	0.27	0.00	0.00	0.00	0.00	0.00	0.00
				MAX		21.68	4.04	56.67	10.56	4.83	1.04	7.22	3.87	6.94	0.00	5.60	0.00
				SD		8.54	1.56	20.11	2.24	1.21	0.25	2.45	1.23	2.52	0.00	2.07	0.00
				CV%		127.91	146.64	83.33	36.41	42.72	39.37	85.56	97.87	128.61	0.00	115.38	0.00
Clone 4				$\overset{–}{X}$		10.72 a	1.59 a	23.52 a	6.00 a	5.38 a	0.64 a	4.28 a	2.76 a	2.25 a	0.20 ab	3.09 a	0.00 a
				MIN		0.00	0.00	7.40	4.37	3.66	0.29	0.00	0.00	0.00	0.00	0.00	0.00
				MAX		27.83	4.40	50.32	8.86	8.09	1.09	8.62	6.17	5.19	1.99	10.03	0.01
				SD		11.10	1.58	18.56	1.43	1.33	0.27	2.74	2.43	1.83	0.58	4.09	0.00
				CV%		103.58	99.09	78.91	23.88	24.69	42.54	63.99	88.04	81.30	286.29	132.06	412.31
ANOVA ^2^				F		0.62	0.47	0.26	0.04	14.67	0.35	2.87	4.01	0.04	2.69	1.60	0.94
				*p*		0.6041	0.7008	0.8561	0.9897	**0.0000**	0.7876	**0.0432**	**0.0111**	0.9881	0.0538	0.1969	0.4277
CDA		Eigenvalues		Percentage		Eigenvalues											
		Z	K	Z	K												
	DA1	200.03	25.57	84.97	67.08	0.24	0.28	**0.90**	0.08	−0.10	**1.02**	0.03	−0.32	0.45	0.00	0.01	0.00
	DA2	26.61	6.35	11.30	16.67	0.12	**0.82**	−0.02	0.61	**1.01**	0.29	−0.13	−0.39	−0.02	0.66	−0.18	0.00

^1^ Mean values with different letter designations within a column are significantly different from each other at the 95% confidence level. ^2^ Variables with discriminant values > 0.70 (CDA) and with *p* ˂ 0.05 (ANOVA) are marked with bold numbers.

## Data Availability

All data are included in this manuscript.

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
