# Peer review of "Phytoremediation Potential of Different Genotypes of Salix alba and S. viminalis"

_plants, 2024, doi:10.3390/plants13050735_

Round 1

Reviewer 1 Report

Comments and Suggestions for Authors

Experimental evidence is to be documented with relevant photo plates. 

Author Response

Dear reviewer,

We are thankful for your suggestions.

Below are our responses.

Reviewer's comment

Our response

Experimental evidence is to be documented with relevant photo plates. 

We thank the Reviewer for this constructive suggestion. We included the requested photo plates as Figure 4 (page 12).

Note. All changes in the manuscript are highlighted in green.

Kind regards,

The authors

Reviewer 2 Report

Comments and Suggestions for Authors

The paper titled "Phytoremediation Potential of Different Genotypes of Salix 2 alba and S. viminalis" addresses a very important and interesting topic. The phytoremediation potential of the genus Salix is known, as pointed out by the authors, and it is interesting to observe this potential in different genotypes. 

The work is interesting, however, it is not an easy read and there are many aspects in the text that can be improved.

It is suggested to accept the job after corrections of minor methodological errors and major changes in the text.

Below are some suggestions:

Line 57: physico-chemical technologies.

Figure 1: One could improve the quality of the image and insert the label on the y-axis with the sibolus of the metal under consideration. It will make it easier to read.

line 136: specify the amount of solution added and the concentrations

lines: 113-117, 121,123, 148-161, 163-170, 171-174, 219-229

Figure 2: Check for typos especially (;). Explain better it is difficult to read and also include details in graphs e.g., units of measurement on y-axis.

Check throughout the text and tables for the decimal separator. In some cases there is a comma in others a dot.

It is recommended that the results of CDA be discussed 

Author Response

Dear reviewer,

We are thankful for your suggestions.

Below are our responses.

Reviewer's comment

Our response

Line 57: physico-chemical technologies.

We altered this term, as suggested (line 57).

Figure 1: One could improve the quality of the image and insert the label on the y-axis with the sibolus of the metal under consideration. It will make it easier to read.

In line with this suggestion, we inserted chemical symbols and measuring units on y-axes to make the graphs easier to interpret (Figure 1).

Line 136: specify the amount of solution added and the concentrations.

This information is given in the Materials and Methods chapter (lines 430–432), in accordance with the journal’s instructions for authors.

Lines: 113-117, 121,123, 148-161, 163-170, 171-174, 219-229

These parts of the text have been changed to make them more clear and easier to read (lines 113–118, 120–125, 157–172, 174–187, 231–241).

Figure 2: Check for typos especially (;). Explain better it is difficult to read and also include details in graphs e.g., units of measurement on y-axis.

We thank the Reviewer for these corrections. We corrected the typos in the caption of the figure and included chemical symbols and measuring units on y-axes to make the graphs easier to interpret (Figure 2).

Check throughout the text and tables for the decimal separator. In some cases there is a comma in others a dot.

We thank the Reviewer for spotting these typos. We checked the text thoroughly to make sure the decimal separator is a dot in all cases.

It is recommended that the results of CDA be discussed.

The results of CDA are discussed in detail in the Results section (lines 249–280) and the Discussion (lines 407–412).

Note. All changes in the manuscript are highlighted in green.

Kind regards,

The authors

Reviewer 3 Report

Comments and Suggestions for Authors

I have gone through the manuscript entitled "Phytoremediation Potential of Different Genotypes of Salix alba and S. viminalis" and found interesting. However, before reaching to the final decision, the below listed point need to be clarify.

1. Authors suggested the variability among the clone but logic behind this is unclear as clone do not varied genetically. how the varied phytoremediation performance among the genetically similar plant material is possible?  It is necessary to clarify the outcome is because of the experimental or statistical  design error. 

2. The mechanism of phytoremediation among the genetical similar material should be the same. Thus the quantity should not be statistically differ. 

3. In the light of the above observation, it is suggested to perform the data analysis again after removing the outlier data points. 

Author Response

Dear reviewer,

We are thankful for your suggestions.

Below are our responses.

Reviewer's comment

Our response

Authors suggested the variability among the clone but logic behind this is unclear as clone do not varied genetically. how the varied phytoremediation performance among the genetically similar plant material is possible?  It is necessary to clarify the outcome is because of the experimental or statistical  design error. 

We are thankful for the concern the Reviewer raised. However, there is no experimental or statistical design error. All data and procedures have been double-checked. Moreover, the obtained results are in line with numerous references cited in the paper (e.g.,  references 17,19,25–31).

The mechanism of phytoremediation among the genetical similar material should be the same. Thus the quantity should not be statistically differ. 

We are thankful for this concern. Still, there is no doubt that different clones may differ in the phytoremediation mechanism, as indicated by many studies published previously (e.g., references 17,19,25–31).

In the light of the above observation, it is suggested to perform the data analysis again after removing the outlier data points. 

As suggested, the data analysis has been performed again, excluding outliers. The results slightly changed and some parts of the manuscript have been rewritten accordingly (lines 113–118, 120–125, 157–172, 174–187, 195, 216, 223, 229, 231–241, 247–252, 267, 337, 350, 361–362, 368–369, 386–387; Tables 2 and 3; Figures 2 and 3). However, the outcome overall is the same as before. The analyzed willow clones statistically differ in terms of their ability to absorb and accumulate different heavy metals, confirming studies exhibiting similar results (e.g., references 17,19,25–31).

Note. All changes in the manuscript are highlighted in green.

Kind regards,

The authors

Round 2

Reviewer 3 Report

Comments and Suggestions for Authors

It is suggested that authors should clearly spell out the reason for varied performance of clone in bioaccumulation of heavy metals.

Author Response

Dear Reviewer,

Thank you for your comments. Please, find our response below.

Reviewer's comment

Our response

It is suggested that authors should clearly spell out the reason for varied performance of clone in bioaccumulation of heavy metals.

We thank the Reviewer for this most constructive suggestion. Based on an additional literature reference, we included a discussion on this matter (lines 380–382, page 10), and improved the conclusion section (line 506, page 13).

Note. All changes in the manuscript are highlighted in green.

Kind regards,

The authors